# Pancreatic Pseudocysts: Evolution of Treatment Approaches

**DOI:** 10.3390/jcm14176152

**Published:** 2025-08-30

**Authors:** Paulina Kluszczyk, Aleksandra Tobiasz, Adam Madej, Piotr Wosiewicz, Sławomir Mrowiec, Beata Jabłońska

**Affiliations:** 1Student Scientific Society, Department of Digestive Tract Surgery, Faculty of Medical Sciences in Katowice, Medical University of Silesia, 40-752 Katowice, Poland; tobiasz.aleksandra00@gmail.com (A.T.); adam.madej22@gmail.com (A.M.); 2Department of Gastroenterology and Hepatology, Faculty of Medical Sciences in Katowice, Medical University of Silesia, 40-752 Katowice, Poland; 3Department of Digestive Tract Surgery, Faculty of Medical Sciences in Katowice, Medical University of Silesia, 40-752 Katowice, Poland

**Keywords:** pancreatic pseudocysts, PPC cystogastrostomy, cystojejunostomy, cystoduodenostomy, laparoscopy drainage, endoscopic drainage, metal stents, plastic stents

## Abstract

Pancreatic pseudocysts (PPCs) are frequent complications of acute and chronic pancreatitis, characterized by encapsulated collections of pancreatic fluid. Historically managed by open surgical approaches, treatment paradigms have significantly evolved with advancements in imaging and minimally invasive techniques. This review outlines the historical progression and current standards in PPC management, covering conservative, surgical, laparoscopic, and endoscopic interventions. Conservative management remains a valid first-line option for asymptomatic, stable pseudocysts, particularly in the absence of complications. Surgical techniques, once the mainstay, such as marsupialization and internal drainage procedures (cystogastrostomy, cystojejunostomy, and cystoduodenostomy), now serve as alternatives when less invasive methods fail. Laparoscopic approaches offer reduced morbidity and faster recovery, especially for complex or inaccessible PPCs. However, endoscopic drainage, particularly endoscopic ultrasound-guided transmural drainage using plastic or metal stents—especially lumen-apposing metal stents (LAMSs)—has become the preferred modality due to its efficacy, safety profile, and cost effectiveness. Emerging technologies, including robotic-assisted surgery and hybrid techniques, promise further refinement in PPC management. This review synthesizes current evidence and expert guidelines, providing a comprehensive overview of evolving strategies and future directions in the treatment of PPCs.

## 1. Introduction

A pancreatic pseudocyst (PPC) is a common local complication of acute (10% to 26%) or chronic (20% to 40%) pancreatitis [1]. According to the latest revision of Atlanta Classification from 2012, PPCs represent encapsulated collections of pancreatic fluid, typically surrounded by a well-defined inflammatory wall of connective tissue outside the pancreas with no or minimal necrosis [2]. The definition of PPC has evolved over time and was not always as clearly delineated. Initially described by Morgagni in 1761 [3], all fluid collections within the peripancreatic region or the pancreatic parenchyma were historically classified under the broad term of PPC [4]. However, advancements in diagnostic imaging, particularly the widespread use of high-resolution computed tomography (CT), magnetic resonance imaging (MRI), and endoscopic ultrasound (EUS), have enabled more precise characterization of PPCs and their relationship to surrounding structures and make their diagnosis easier.

Since the 18th century, PPCs have been the focus of extensive investigation due to their varied clinical presentations and potential complications, including infection, hemorrhage, and rupture [5,6,7]. Historically, the interventional management of PPCs was limited to surgical procedures, often associated with significant complications and prolonged recovery. However, over the years, surgeons have developed new surgical techniques to treat PPCs and improved existing ones. Simultaneously, the development of minimally invasive techniques has transformed the therapeutic landscape, offering effective alternatives to surgery with reduced patient burden [8,9].

This review explores the evolution of treatment approaches for PPCs, emphasizing the shift from open surgical drainage to minimally invasive strategies, including endoscopic and laparoscopic techniques. It also aims to provide a detailed analysis of past and current treatment strategies while highlighting emerging approaches that are redefining standards of care and improving patient outcomes.

## 2. Conservative Treatment

Conservative management remains a widely accepted and currently preferred approach in the treatment of stable, non-enlarging PPCs, particularly in asymptomatic patients [10,11]. Recent studies demonstrate its safety and efficacy in enabling spontaneous resolution of PPCs in 86% of cases at 1-year follow-up, with a major complication rate of 3% to 9%. This represents a significant contrast to earlier reports from the 1970s, in which a follow-up period of more than seven weeks was associated with a higher risk compared to elective surgical intervention, showing only a 20% spontaneous resolution rate for PPCs [1].

After supportive medical care, spontaneous resolution of cysts that have developed after acute pancreatitis is more common in contrast to their spontaneous resolution after an episode of chronic pancreatitis [10]. Available data suggest that non-operative management leads to cyst resolution in 57% of patients, with overall resolution reported in up to two-thirds of cases following cyst maturation [11,12].

Initially, PPCs less than 4 cm were considered a factor for spontaneous resolution [13], with a later cut-off factor revised to <6 cm [14]. However, subsequent evidence has shown that the size of the PPC alone is not a reliable predictor of spontaneous resolution, need for intervention, risk of recurrence, complications, or mortality [1,15]. Nevertheless, larger PPCs are more frequently associated with symptomatology and complications, key indications for drainage treatment. Complications may include hemorrhage, obstruction, compression of the pancreatic or biliary ducts, or infection of the cyst [11,12]. Additional factors that reduce the chances of successful conservative management include PPCs located in the tail of the pancreas and the presence of multiple cysts [10].

The conservative approach typically involves imaging monitoring for changes in cyst size or the emergence of complications in addition to supportive care, including analgesics and antiemetics. Dietary modifications, particularly adherence to a low-fat diet, are also recommended [10,12].

Several case series have also described the use of the somatostatin analogue octreotide, which shows a marked inhibitory effect on pancreatic exocrine secretion [16,17,18]. In one study, octreotide administered at 100 µg subcutaneously three times daily for two weeks was associated with clinical improvement and reduction in cyst size in four out of seven patients [16]. Other reports suggest that the concurrent use of octreotide, administered subcutaneously in doses of 50–1000 µg three times a day, may shorten the duration of percutaneous drainage [19].

A recent case described successful management of PPCs with atypical hepatic localization in an 8-year-old girl using a combination of octreotide and intercostal drainage, with favorable outcomes [20].

## 3. Interventional Management

Since the 19th century, various surgical approaches have been developed for the treatment of PPCs. This section explores these approaches, discussing their origins, mechanisms, and effectiveness, as well as their historical significance and current relevance. The most important dates in the history of PPC treatment are presented in Figure 1. Additionally, the interventional methods of treatment in PPCs are presented in Figure 2.

## 4. Surgical Treatment

### 4.1. Percutaneous Drainage (PCD)

Percutaneous surgical drainage (Figure 3) was first performed by A. Le Dentu in 1865, marking one of the earliest surgical interventions on the human pancreas. It can be performed through either a simple percutaneous aspiration or percutaneous catheter drainage guided by ultrasound sonography (USG) or CT. In the latter approach, a pigtail catheter is inserted into the PPC over a guidewire placed with needle assistance [21]. Catheter sizes used for draining PPCs typically ranged from 8 to 14 F, with larger sizes being more commonly utilized for infected PPCs. In a series of 101 cases, mean catheter duration was 16.7 days for infected and 21.2 days for noninfected pseudocysts [22].

The initial method, percutaneous aspiration, for the first time under USG guidance was performed in 1976 by Hancke and Pedersen. It showed promising results in preliminary studies conducted in the 1970s and 1980s [23,24]. However, subsequent observations revealed a recurrence rate exceeding 70% and a failure rate of 54%, along with increased complications from repeated aspirations. It is therefore not recommended in current practice [8,21,25].

In contrast, percutaneous catheter drainage under USG or CT guidance has demonstrated significantly higher effectiveness. Even before the era of USG and CT, external drainage by a catheter was found to be as effective as marsupialization, with significantly lower morbidity rates [26]. Studies from the late 20th century reported 85–96% success, 10–20% recurrence, and 1–6% mortality [5,22,25,27]. More recent data are less favorable: a 2021 meta-analysis of 1398 patients with PPCs or pancreatic necrosis found 63% clinical success (defined as improvement and reversal of organ failure) and 13% mortality, and also less than one-third of patients required surgical intervention after the percutaneous drainage procedure [28]. This method is also not free from complications, the most common of which are the development of a fistula, hemorrhage, sepsis, and infection [28,29]. Some of them may be associated with prolonged catheter placement. To reduce the duration of drainage, an adjunctive therapy with octreotide—a somatostatin analog that reduces pancreatic secretions through multiple pharmacologic mechanisms—had been explored. Clinical evidence from a study involving seven patients suggested that octreotide may effectively reduce catheter output, potentially shortening drainage duration and mitigating associated complications [19].

While percutaneous surgical drainage is still utilized, it is considered less favorable than surgical or endoscopic procedures [9,29,30]. Its application is now largely limited to specific cases, such as patients with high surgical risk, who cannot tolerate other endoscopic or surgical procedures, patients with immature or infected PPCs, or those requiring immediate pseudocyst-related symptom relief, particularly for pain [21,24,29,31]. This may partly explain the declining outcomes of percutaneous drainage, as patients selected for this treatment likely exhibit poor prognostic criteria for surgical or endoscopic interventions [29].

### 4.2. External Surgical Drainage: Marsupialization

The first documented preoperative diagnosis of a PPC, leading to planned pancreatic surgery, was made by Carl Gussenbauer in 1882, who treated it with marsupialization (Figure 4) [32]. In this open procedure, an incision is made in the PPC, and its edges are sutured to the parietal peritoneum, creating a continuous surface with the abdominal cavity to enable drainage and prevent closure. This technique remained the gold standard for PPC surgery until the 1920s, when more effective methods emerged [33,34].

Brandenburg et al. (1951) identified the main drawbacks of marsupialization as persistent fistulas, premature sinus tract closure with cyst recurrence, skin excoriation, and infection or suppuration of the cyst cavity [35]. Recurrence was common and often required reoperation, though mortality remained relatively low (4–6%) [36]. Today, marsupialization is rarely used but may be valuable in select cases, such as hemorrhagic cysts, where both fluid and blood clots must be evacuated [37].

### 4.3. Internal Surgical Drainage: Cystogastrostomy, Cystojejunostomy, and Cystoduodenostomy

In 1911, Louis Ombrédanne performed the first pancreatic cystoduodenostomy, creating an anastomosis between the duodenum and a PPC to drain its contents into the gastrointestinal tract (Figure 5A) [33]. Although the patient died 11 days postoperatively, the case spurred further research into internal anastomosis for PPC treatment. In 1923, Rudolf Jedlicka reported the first case of anastomosis between a PPC and the stomach, in which most of the PPC was resected, and the remnants were connected to the posterior gastric wall [33]. His operation is considered to be the first pancreatic cystogastrostomy. Antoni Jurasz refined the approach in 1929 by introducing a transgastric technique: opening the anterior stomach wall, puncturing and evacuating the PPC via the posterior wall, and creating a wide anastomosis between the cyst and stomach (Figure 5B) [38,39]. Because of its minimal postoperative complications, this technique became the preferred method of treatment of PPCs adhered to the gastric wall [40,41]. Rare but possible complications of this surgery include bleeding from gastric mucosa, bleeding from the cyst, or mechanical ileus [42,43].

In 1892, César Roux performed the first Roux-en-Y gastrointestinal reconstruction for the treatment of anthropyloric obstruction [44]. However, it took over 50 years until the Roux-en-Y technique was implemented in the treatment of PPCs. Adolf Henle performed the first pancreatocystojejunostomy in 1923 [45], and in 1946, E. Köning combined it with the Roux-en-Y configuration [33].

A Roux-en-Y pancreatocystojejunostomy (Figure 5C) is an internal surgical drainage procedure that involves creating an anastomosis of the PPC and the small intestine. The first loop of the jejunum is dissected and transected, and then a Y-shaped configuration is created; the side-to-side anastomosis is performed between the closed jejunal loop and the cyst wall, and jejuno-jejunal end-to-side anastomosis approximately 50 cm distally to the cystojejunostomy is performed.

For many years Roux-en-Y cystojejunostomy was the first-choice technique for PPC treatment due to its high effectiveness and low number of complications [46,47]. Potential complications may include postoperative bleeding, wound infection, and anastomosis leakage [43,47,48].

When comparing cystogastrostomy and cystojejunostomy for the treatment of PPCs, both procedures demonstrate similar mortality rates. However, limited comparative studies suggest that cystogastrostomy is associated with higher morbidity and recurrence rates compared to cystojejunostomy. Notably, postoperative gastrointestinal bleeding occurs more frequently after cystogastrostomy, likely due to the gastric wall’s rich vascularity [43,47,48].

Cystoduodenostomy, though historically the first surgical procedure developed for internal drainage of PPCs, is now rarely performed, reserved for PPCs in the pancreatic head abutting the duodenal wall [47]. Its technical difficulty, with the risk of injuring the gastroduodenal artery or common bile duct, may explain its decline [49]. Complications include fistulas, bleeding, wound infection, and postoperative pancreatitis, with a recurrence rate of approximately 5% [47,50].

For many years, internal surgical drainage was the standard treatment for PPCs. While surgical approaches are increasingly being replaced by less invasive techniques such as laparoscopy and endoscopy, certain scenarios still necessitate surgery. These include recurrent PPCs, PPCs that are challenging to access endoscopically, or those with uncertain etiology. Given the limited comparative studies on comparison of different internal surgical drainage methods, the choice between cystojejunostomy, cystogastrostomy, and cystoduodenostomy should be based on the PPC’s location and size and the surgeon’s expertise [10,21,47,51].

## 5. Laparoscopy Drainage

In 1994, the first laparoscopic cystojejunostomy was performed by Constantine T. Frantzides in a 34-year-old patient after a second incident of a PPC caused by alcohol abuse. Due to its unfortunate location, posterior to the transverse colon, a percutaneous method could not be performed. During surgery, after placement of the trocars and exposing the base of the cyst, it was punctured and injected with contrast. The PPC, with its extent, was confirmed by fluoroscopy, and due to its location, the proximal part of the jejunum was chosen as best placement of the anastomosis to the PPC. After anchoring the jejunum to the inferior wall of the PPC, an incision was made, and the internal part of the PPC as well as stapled anastomosis were inspected with a laparoscope; subsequently, a 6 cm long transverse skin incision was made in the left upper quadrant, and the jejunum proximal and distal to the anastomosis was handsewn side-to-side to prepare for enteroenterostomy. After reducing the intestine into the abdomen, the skin incisions were closed [52]. It was the first surgery of its kind and the first partially laparoscopic method of PPC treatment.

### There Are Three Categories of Laparoscopic Approaches for PPCs: Laparoendoscopic, Extragastric, and Roux-en-Y Cystojejunostomy

The laparoendoscopic approach is very similar to endoscopic transmural internal drainage, but it is allegedly safer due to additional laparoscopic trocars. The base working principle of laparoendoscopic treatment is to place transabdominal trocars into the stomach, which offers much better visualization of access, incision and drainage of the pseudocysts, and thus a safer and more accurate method. Minilaparoscopic cystic gastrostomy, however, is a method that includes using a gastroscope for easier visualization and aspiration of PPC contents as well as minilaparoscopic instruments, which are used for creating a posterior gastric wall incision and cyst gastrostomy [51,53]. The main indication for such an approach is the large size of a PPC, as small PPCs will only achieve minor communication with the stomach, as the area of contact is relatively small between them; thus, drainage will not be sufficient enough [54].

Laparoscopic extragastric cystogastrostomy can be performed only if the anterior wall of the PPC is in direct proximity to the posterior wall of the stomach, since it involves creating a connection between the PPC and lumen of the stomach with a stapler, through which contents of the PPC can be drained [55,56]. The operation involves creating a CO_2_ pneumoperitoneum and placing a standard port and laparoscope into the peritoneal cavity; then, through a nasogastric tube, the stomach needs to be insufflated with CO_2_ while reducing intraabdominal pressure. Subsequently, an intraluminar 5 mm trocar is inserted into the abdominal cavity and stomach through its anterior wall under direct vision, and after thorough visual confirmation of good positioning, a second trocar is inserted the same way several centimeters from the first one, and additional trocars can be inserted if needed the same way. When all trocars are positioned, the peritoneal cavity is evacuated completely, the stomach can be insufflated with the maximum size, and cystogastrostomy can be performed [56]. A relatively small PPC can be drained by that method via a lesser sac approach, as this way, much larger anastomosis can be achieved than by the transgastric method [54].

Roux-en-Y cystojejunostomy was a method preferred for PPCs located in a more caudal position. It is a rather rare surgical method, as its indications are quite uncommon: a big PPC in the form of a bulge that is located posterior to the colon [54]. After opening an omentum, it is necessary to locate the PPC using an ultrasound or aspiration method; then, the proximal loop of the jejunum is transected with a linear stapler, and cystojejunostomy with a distal part of the jejunum is being made with a linear stapler. Subsequently, the Roux-en-Y is performed by stitching the bowel connected via cystojejunostomy to the distal part of the previously dissected jejunum [57].

An alternative method of cystojejunostomy is the laparoscopic loop cystojejunostomy, in which a cystostomy is created between the PPC and a loop of the jejunum, without the need for Roux-en-Y loop reconstruction. This procedure can reduce the operation time; however, it is rarely used due to the risk of anastomotic dehiscence and the potential reflux of intestinal contents into the cyst [58,59].

A general laparoscopic approach of PPCs, no matter its type, usually avoids common shortcomings present for endoscopic and percutaneous drainage, such as insufficient drainage, because it is possible to create a larger cystostomy; tract infections, as there is possibility for concomitant pancreatic necrosectomy; and the obstruction of small cystogastrostomy, due to the fact that it is possible to form a larger enterostomy via a smaller sac [60].

Although recent clinical experience increasingly supports endoscopic techniques as a routine first-line modality for the management of PPCs, the role of laparoscopic drainage remains significant, particularly in cases where endoscopic access is not feasible or in pseudocysts with complex anatomical locations. In a study by Hamza and Ammori, the safety and efficacy of various laparoscopic drainage techniques were evaluated in PPCs meeting the 1992 Atlanta criteria for surgical intervention [54]. They grouped pseudocysts into five categories and selected different surgical methods accordingly. Large retrogastric PPCs were resolved by transgastric cystogastrostomy; small retrogastric PPCs, located in the splenic hilum and gastrohepatic ligament, by extragastric jejuniostomy; infracolic PPCs byRoux-en-Y jejunostomy; and PPCs located in the pancreatic head by cystoduodenostomy. In this study, of the 30 procedures, only one conversion to laparotomy, a 3.3% morbidity rate, and no mortality were observed. In addition, the recurrence rate was 7% [54].

The outcomes of laparoscopic interventions have been repeatedly assessed in the literature. In 2003, Bhattacharya et al. reported an overall success rate of 89%, with a morbidity rate of 7% and no observed recurrences [60]. A subsequent meta-analysis by Lerch et al. in 2009, which included 14 studies published between 1998 and 2008, demonstrated improved outcomes, with a success rate of 92%, no morbidity, and a recurrence rate of 3% [61]. These data suggest that continuous refinements in laparoscopic techniques have contributed to improved efficacy and safety in PPC management over time.

Moreover, laparoscopic drainage has proven to be a more beneficial approach for treating PPCs, offering superior outcomes compared to both open surgical and percutaneous methods. According to a large population-based study by Wang et al., laparoscopic drainage is associated with the lowest rates of short-term complications, including acute renal failure, sepsis, and respiratory issues. It also results in a significantly shorter hospital stay—on average 4 days less than open surgery and 2 days less than percutaneous drainage—along with lower hospitalization costs. Unlike the open surgical approach, laparoscopic procedures are less invasive, minimize blood loss, and reduce the need for transfusions, making them a safer and more cost-effective option [62].

However, recent comparative studies have questioned whether laparoscopic drainage offers a distinct advantage over endoscopic or open surgical methods in terms of long-term success, morbidity, and mortality. Some reports have suggested minimal differences between these modalities, though minor benefits of endoscopic techniques—such as shorter operative time, reduced blood loss, and briefer hospitalization—have been noted [63,64,65].

A first meta-analysis (from 2021) compared a group of patients treated with endoscopic and laparoscopic methods and presented the following results: treatment success rate (50–100% vs. 78.9–100%) and overall success rate (89.2% vs. 88.8%), showing that there were no differences in treatment rates between the two groups. In addition, the total rate of adverse events was compared, which was 11.35% in the endoscopic group and 14.66% in the laparoscopic group. The rates ranged from 0–21.4% and 8.3–26.3%, respectively. No significant differences were observed here either.

However, significant differences were found in the duration of surgery and hospitalization, showing that they were significantly shorter and with less blood loss in the endoscopic group than in the laparoscopic group. Additional analyses revealed that there was no difference in the recurrence rates between the two groups [64].

It is important to consider that laparoscopic approaches are contraindicated in certain clinical scenarios, including infected or obstructed PPCs, immature cyst walls, and patients who are hemodynamically unstable. In such cases, percutaneous drainage is preferred despite its higher risk of recurrence and potential for pancreatic fistula formation [66,67,68].

Current evidence suggests that endoscopic approaches may be more suitable for managing chronic PPCs within the head and body of the gland, while laparoscopic interventions are better suited for acute PPCs, particularly those associated with necrotizing pancreatitis due to remove more necrotic tissue from the cyst while exploring the cyst wall structure than endoscopic drainage.

The use of laparoscopy is limited by incomplete anastomosis, gastric perforation, or abdominal contamination. Nevertheless, laparoscopic surgery of the pancreas is becoming an increasingly popular method of treating mature PPCs, as it is minimally invasive and provides effective drainage [55,64,65,68].

## 6. Endoscopic Drainage of PPCs

### 6.1. At the Beginning

Following the initial surgical management of PPCs, the introduction of endoscopic drainage marked a significant advancement in minimally invasive therapy. The primary goal of endoscopic drainage is to establish a fistulous tract between the cystic cavity and the lumen of the gastrointestinal tract (duodenum or stomach), enabling internal drainage of the cystic contents [1].

The first report of endoscopic drainage dates back to 1975, when B.H. Gerald Rogers performed a transgastric needle aspiration of a pseudocyst via an endoscope, utilizing a biopsy channel [69]. In 1983, Khawaja and Goldman described the first successful endoscopic cystogastrostomy (ECG) [51,70], and in 1985, R.A. Kozarek pointed to the potential of the procedure in patients with high surgical risk or failed prior drainage attempts [71]. Between 1981 and 1985, J. Sahel performed 20 endoscopic cystostomies with a 90% success rate, using Olympus endoscopic instruments and a custom-made diathermic knife.

Initially, this method was recommended only for pseudocysts in close proximity to the gastrointestinal wall. However, over time, it has proved to be a viable alternative to surgery, offering comparable mortality and morbidity outcomes, with the additional benefit of avoiding general anesthesia [72].

### 6.2. Technical Developments in Endoscopic Drainage

#### Early Advances in Endoscopic Management of PPCs: Insights from Cremer and Sahel

As early as 1989, Cremer [73] demonstrated that endoscopic cystoduodenostomy (ECD) represented the first-line approach for the management of paraduodenal cysts. ECG was established as an alternative procedure, particularly for PPCs located posterior to the stomach (retrogastric). Endoscopic intervention was indicated only when the cyst was in direct contact with the gastrointestinal wall, at a distance of less than 1 cm. In all patients, CT was performed to assess the size and anatomical position of the cyst.

The initial ECG procedures were carried out using a standard Olympus JF 1T duodenoscope, followed by the Olympus TJF 10 model and, less frequently, a panendoscope. Puncture of the gastrointestinal wall was performed perpendicularly to the closely apposed cyst using a 5-French diathermy needle, which also allowed for contrast injection for accurate fluoroscopic guidance. Following this, a 10-French sheath (later also used with a diathermic tip) was introduced, through which a large catheter was advanced into the cyst; the needle was subsequently withdrawn.

The cystoenterostomy tract was enlarged by coagulation up to 8 mm for duodenal access, up to 10 mm for gastric access, as reported by Cremer (1989) [73], or between 8 and 20 mm, according to Sahel (1991) [74].

Drainage was maintained using a nasocystic catheter: 6 French in duodenal approaches and 9 French in gastric approaches. The catheter remained in place until complete regression of the cyst was achieved. The drainage period varied from 4 to 30 days according to Cremer [73] or 2 to 7 days according to Sahel [74]. When a double-pigtail stent was employed, drainage could be prolonged for 1 to 2 months [74].

In 1994, a novel instrument for transmural puncture of PPCs was introduced. It comprised a diathermic wire in contact with the needle, housed within a retractable inner injection catheter, and encased in an outer 7 F Teflon sheath [75].

### 6.3. A Decade of Progress: Endoscopic Drainage of PPCs (1990–2000)

#### 6.3.1. Early 1990s: Foundations and Initial Outcomes

In the early 1990s, the endoscopic approach to PPC drainage using fluoroscopic control was already showing promising results [73,74]. According to Sahel (1991), the overall success rate of endoscopic treatment was estimated at 90%, with a morbidity rate of 11% and a mortality rate of 3.3% [74]. Significant clinical improvements were observed, particularly in pain relief and reduction of recurrence rates, which were reported as 9% after ECD and 19% after ECG. Nonetheless, several procedure-related complications were documented, including bleeding, retroperitonitis, cyst infection, and retroperitoneal perforation.

By the end of the decade, the number of endoscopic procedures performed had increased substantially, while the mortality rate dropped significantly to 0.23%, markedly lower than that associated with surgical drainage. The overall clinical success rate remained stable at approximately 90%, and the recurrence rate after endoscopic treatment—at around 16%—was comparable to outcomes observed in surgical cohorts [21]. In expert hands, initial technical success was reported to reach as high as 94% [76]. The complication rate was recorded between 16% and 20%, with ECD continuing to demonstrate fewer adverse events than ECG.

Despite these improvements, approximately 17% of patients still required additional non-endoscopic interventions, predominantly surgical procedures, indicating the importance of individualized treatment planning and interdisciplinary care in complex or refractory cases. Nevertheless, the endoscopic method increasingly replaced traditional surgical approaches as the treatment of choice [21,73,76,77].

#### 6.3.2. Transpapillary and Transmural Drainage: Technical Aspects

Over time, endoscopic management of PPCs has evolved into two primary approaches: transpapillary drainage and transmural drainage, with either transduodenal or transgastric access.

Transpapillary drainage involves an initial endoscopic biliary and pancreatic sphincterotomy, followed by balloon dilatation of pancreatic duct (PD) strictures, which are frequently present. Subsequently, stents ranging from 5 F to 10 F in diameter and 3 to 12 cm in length are deployed. These stents are typically replaced every 6 to 8 weeks, with the overall treatment duration ranging from 3 to 8 months and follow-up periods extending from 15 to 37 months [60,78,79].

Outcomes from several case series have reported clinical success rates of 81–94%, recurrence rates around 9%, hemorrhagic complications <1%, and acute pancreatitis incidence of approximately 5%. These results indicate that transpapillary drainage is generally a safe and effective technique [10,78,80,81,82].

However, a 2017 meta-analysis concluded that combined drainage (i.e., transpapillary PD stenting in conjunction with transmural drainage) did not offer additional clinical benefit over transmural drainage alone [83]. As such, transpapillary drainage under endoscopic retrograde pancreatography (ERCP) guidance is now typically reserved for

PPCs smaller than 6 cm;Cases with a documented connection to the main pancreatic duct, observed in 36–69% of patients [78,84];Situations where transmural drainage is contraindicated, such as in coagulopathy or when the distance between the PPC and gastrointestinal lumen exceeds 1 cm [1,10,85].

Transmural Drainage: This approach begins with the identification of the site of maximal bulging, typically using a side-viewing endoscope or with EUS in cases of non-bulging PPCs [60,80,82], which occur in 42–48% of patients [10].

The procedural steps involve the insertion of a needle catheter through the GI wall into the pseudocyst, injection of contrast to confirm appropriate placement, dilation of the cyst-enteric fistula using 8–15 mm biliary balloon dilators, and finally placement of one or more polyethylene stents (7 F or 10 F) into the pseudocyst cavity. These stents may remain in place for 1 to 4 months or until radiological confirmation of resolution is obtained [12,60,80,86]. In cases of persistent cysts after 4–6 weeks, stent replacement is recommended. However, subsequent multivariate analyses identified that implantation of multiple endoprostheses (instead of single stent placement) and drainage duration longer than 6 weeks predict a more favorable outcome of endoscopic drainage. The placement of more than one stent provides a wider drainage opening and a lower probability of stent occlusion, which appears to result in better drainage [86].

Reported outcomes have improved over time. Initial success rates ranged from 36% to 90–97% [60,82,87,88].

Additionally, technical success rates have been associated with

PPCs located in the head or body of the pancreas (compared to the tail);Lesions < 1 cm from the gastrointestinal lumen;Pseudocysts related to chronic or necrotizing pancreatitis rather than acute presentations [60].

The recurrence rate for transmural drainage is estimated at around 5% [82], while for both methods combined (transmural and transpapillary drainage), it is around 8–9% [88].

### 6.4. Stents in Endoscopic Drainage of PPCs

Various types of stents are described in the literature for the management of PPCs, including nasocystic catheters, double-pigtail plastic stents (DPPSs), self-expanding metal stents (SEMSs), combination DPPSs with SEMSs, and lumen-apposing metal stents (LAMSs) [1,89].

### 6.5. Nasocystic Catheters and Plastic Stents

Initially, as mentioned previously, nasocystic catheters or plastic stents (ranging from 7 F to 10 F, depending on the cyst’s contents) were commonly used for endoscopic drainage of PPCs [82]. A study published in 2005 highlighted the benefits of using pigtail stents over straight stents to avoid major complications [86] (Figure 6A–D).

### 6.6. Timing of Stent Removal

A randomized trial in 2007 revealed that early removal of stents after successful transmural drainage of PPCs was associated with a higher recurrence rate compared to leaving the stent in place. This approach was considered beneficial for maintaining the patency of the cystoenterostomy formed during the drainage process [90]. This finding challenged previous studies that recommended early removal of stents after PPC resolution to avoid stent occlusion or reduce the risk of infection due to the foreign nature of the stents [90,91]

In cases of large or infected PPCs (containing purulent or necrotic material), nasocystic catheters were used to flush the cyst before inserting one or more stents for effective drainage [82,90].

### 6.7. Self-Expandable Metal Stents (SEMSs)

Introduced in 2010, SEMSs became the standard for endoscopic drainage for some time (Figure 7A,B). Compared to earlier used plastic stents, which often required multiple placements due to their smaller lumen diameter, SEMSs offer a much larger lumen and therefore improved drainage efficiency of PPCs [1,92]. However, SEMSs have been associated with a higher risk of stent migration. To address this, one study reported that placing a DPPS inside a fully covered SEMS (FCSEMS) eliminated stent migration in all cases [92].

Further studies demonstrated that FCSEMSs had a significantly higher resolution rate (98%) for PPC drainage compared to DPPSs (89%). Additionally, SEMSs were associated with fewer procedural adverse events (16%) compared to DPPSs (31%) [93].

### 6.8. Lumen-Apposing Metal Stents (LAMSs)

Around 10 years ago, a new type of prosthesis, the lumen-apposing metal stent (LAMS), was developed and has since been compared with both DPPSs and SEMSs in numerous studies [11,89,94]. LAMSs are flexible, fully covered, self-expanding metal stents with a diameter of 10/12/14/16 mm, providing a larger caliber than DPPSs and offering improved drainage efficiency. They are designed with wide flanges at both ends, which help approximate the gastrointestinal lumen to the PPC wall and reduce the risk of stent dislocation [1,11]. The review study from 2019 shows that the average time for LAMS removal was between 8 and 12 weeks [95].

A meta-analysis of 933 patients demonstrated 98% clinical success and 97% technical success in PPC drainage using LAMSs. High success rates were also noted for the drainage of walled-off pancreatic necrosis (WOPN) and gallbladder (GB) conditions [96].

### 6.9. LAMS- and DPPS-Related Complications

Complications related to LAMSs occur in approximately 0–41.9% of cases and related to DPPSs occur in 5–48.4% of cases and include issues such as stent migration, bleeding, infection, perforation of the PPC, pneumoperitoneum, and abdominal or back pain [11,97,98,99]. The complication rate in patients treated with LAMSs was lower than in the DPPS group (16% vs. 20.2%). The most common complication in the LAMS group was bleeding, whereas in the DPPS group, it was infection [99]. However, another study shows that infection rate (bacterial and fungal) was higher in the LAMS group compared to DPPSs but is equally high and higher for all patients with WON regardless of the stent group. Additionally, procedure- and disease-related complications occurred in nearly all patients with WON (95%) compared to only half of the patients with PPCs (50%) [100]. Furthermore, the stent migration rate was lower in the LAMS compared to the DPPS group (0.9% vs. 2.2%) [99].

A randomized trial examining limited necrosis drainage involving LAMSs suggested that stent removal should occur within 3 weeks, as prolonged stent placement increases the risk of adverse events, especially bleeding. For long-term drainage, replacing a LAMS with a DPPS may be a viable solution [101]. The mortality rate was similar in both groups of stents [99].

### 6.10. Clinical Success and Recurrence

Compared to DPPSs, LAMSs are generally used for larger PPCs, have a shorter duration of placement, and are associated with fewer recurrences. Studies comparing both stent types showed that LAMSs had a higher clinical success rate (96% vs. 87%), shorter procedure times, and fewer percutaneous interventions [102]. However, in other studies, both LAMSs and DPPSs demonstrate a similar high therapeutic success rate [99,103].

A 2018 meta-analysis revealed a higher clinical success rate and lower incidence of adverse effects for metal stents compared to plastic stents in the endoscopic treatment of PPCs [94]. Additionally, a selective approach combining LAMSs with plastic stents yielded significantly better treatment success compared to using only plastic stents [104].

A multicenter study, WONDER-02, was designed to evaluate the non-inferiority of plastic stents compared to LAMSs in patients with PPCs. The study noted that in cases of liquid-filled PPCs, they often resolve without the need for more expensive and potentially riskier LAMSs. The authors also pointed out the lack of solid evidence supporting the superiority of LAMSs over DPPSs, despite their frequent use. Therefore, while LAMSs are increasingly popular and available, the choice of stent should be individualized for each patient by an experienced endoscopist [105].

### 6.11. Imaging Evaluation in Endoscopic Drainage

As early as 1992, H. Grimm highlighted the limitations of the ‘blind’ approach in earlier endoscopic techniques for PPC drainage. The primary challenge was the difficulty in accurately puncturing the pseudocyst without direct endosonographic guidance, which posed significant risks of bleeding and perforation. Grimm demonstrated the critical role of EUS in identifying the optimal puncture site for PPCs, particularly when the cysts did not exhibit extramural bulging [106].

While transmural endoscopic drainage without ultrasound guidance has been described, such methods were less precise. One example includes diathermy puncture using fluoroscopy [107] and another is the Seldinger technique, which involves puncturing the cyst with contrast injection, followed by placement of a guidewire and catheter under radiographic confirmation [108].

### 6.12. Endoscopic Ultrasound (EUS)

EUS has become an integral part of endoscopic procedures, enabling accurate localization of PPCs and measurement of the distance between the cyst and the mucosal layer, especially in cases where intraluminal impressions of the PPC are not well visualized (Figure 8A,B). EUS also serves a vital role in identifying submucosal vessels, pseudoaneurysms, and other vascular structures, all of which contribute to minimizing the risk of bleeding during the procedure [60,76]. Furthermore, EUS is invaluable in diagnosing cystic neoplasms or other cystic lesions of the pancreas [76,109].

A 2007 international survey of 266 endoscopists reported that CT imaging was used by 95% of participants before endoscopic drainage. However, EUS was employed by 70% of US-based endoscopists and 59% of international endoscopists before drainage. Furthermore, EUS-guided drainage was performed by 56% of US and 43% of international practitioners, respectively [82]. A 2008 study reported a higher technical success rate when EUS was used compared to when it was not (95.6% vs. 59.1%) [110].

Over the past decade, EUS-guided endoscopic transmural drainage has become the first-line treatment approach for managing persistent and symptomatic PPCs, including both sterile and infected variants. When compared with percutaneous or surgical drainage modalities, EUS-guided approaches demonstrate comparable technical success while offering a significantly lower incidence of procedural morbidity, reduced need for repeat interventions, and a shorter duration of inpatient care [101].

### 6.13. Endoscopic Retrograde Pancreatography (ERCP)

Another imaging modality, ERCP, is particularly useful in demonstrating the communication between the pseudocyst and the pancreatic duct (PD). However, given the availability of other imaging techniques, ERCP is not routinely required in most cases [60,78,85].

### 6.14. Transabdominal Ultrasound and CT Imaging

Transabdominal ultrasound remains an important imaging technique, with a sensitivity of 75–90% in detecting hypoechoic or anechoic PPCs. Doppler ultrasound is particularly useful for excluding pseudoaneurysms. However, due to the potential for increased intestinal gas interfering with visualization, contrast-enhanced CT is considered the preferred imaging study in the acute setting (Figure 9). CT has the advantage of being able to visualize additional pathologies around the PPC and offers a sensitivity range of 90–100% for identifying PPCs and related complications [85].

### 6.15. Magnetic Resonance Imaging (MRI) and Magnetic Resonance Cholangiopancreatography (MRCP)

MRI or MRCP may offer superior sensitivity and visualization compared to CT or ultrasound, particularly when assessing the presence of PPCs, their contents, solid debris, the patency of transmural drainage, or the presence of hemorrhage within the PPC [85,90].

### 6.16. Antibiotic Prophylaxis and Infections in Endoscopic Drainage

The role of antibiotics during the perioperative period has been widely debated. Some authors advocate for their routine use for prophylaxis, with commonly recommended regimens including ampicillin sodium–sulbactam sodium or vancomycin and gentamicin for patients with penicillin allergies [78,86]. In a large study, fluoroquinolone was administered for three days after the EUS procedure, leading to a very low infection rate: only one infection was reported out of 603 patients. In total, 90% of the patients received antibiotic therapy; among them, 2.4% had complications, and their average pancreatic cystic lesion size was larger (26.4 ± 0.8 mm), while the 60 patients who did not receive prophylactic antibiotics had no complications, and their average cyst size was smaller (19.4 ± 1.5 mm) [111].

Conversely, other authors recommend the use of antibiotics only when infection is established [60,112].

One study shows that hospital-acquired infections occurred in 86% of all patients (with LAMSs, DPPSs, and both stents) and comprised both general infectious complications, specifically disease- or procedure-associated causes. In addition, patients with WON have a higher rate of bacterial and fungal infections by relatively low values in patients with PPCs. Almost equal proportions of Gram-positive and Gram-negative bacteria were found, and in most cases, facultatively anaerobic species were identified as the leading pathogen. About two-thirds of all antibiotic agents used were penicillin derivatives and carbapenems. Nearly half of the DPPS patients with WON had a monoinfection with Candida albicans, whereas none of the PPC patients in the same group were affected (47% vs. 0%). Caspofungin was the most commonly administered antifungal agent, followed by fluconazole and voriconazole [101].

## 7. Future Perspectives and Technical Modifications

### 7.1. Robotic Surgery

Robot-assisted surgery has emerged as a promising approach for the treatment of PPCs, particularly in cases requiring surgical drainage when endoscopic or percutaneous methods are unsuitable. Using systems such as the Da Vinci Surgical System, surgeons can perform procedures including cystogastrostomy or Roux-en-Y cystojejunostomy with enhanced dexterity and visualization. These platforms provide high-definition, three-dimensional imaging, and articulated instruments, enabling meticulous dissection and intracorporeal suturing in the anatomically challenging pancreatic region [113,114,115,116,117,118].

Several case reports and series have demonstrated the feasibility and potential benefits of this approach. Marino et al. reported on 14 patients undergoing robotic-assisted drainage of PPCs, achieving a primary success rate of 85.7%, low major morbidity (14.3%), and no 30-day mortality. Patients also experienced shorter hospital stays and faster postoperative recovery when compared to traditional open surgery [115]. Similarly, Neri et al. described a case of robotic transgastric cystogastrostomy for a giant 15 cm PPC with postoperative stability and resolution of the lesion. The robotic platform facilitated precise anastomosis and allowed for subsequent endoscopic debridement of residual necrosis [117].

A distinct technical variation, robotic retrogastric cystogastrostomy, has been described by Felsenreich et al. This approach avoids anterior gastrotomy, requiring only a single posterior gastric incision to access the PPC. It offers improved visualization with limited dissection and can be combined with procedures such as cholecystectomy, as demonstrated in their case of gallstone-induced PPCs [118].

Robotic surgery may also be particularly advantageous in anatomically complex or urgent cases. Broquet et al. reported the successful emergency use of robotic Roux-en-Y cystojejunostomy in a patient with a large infected PPC and concurrent hiatal hernia, which precluded safe endoscopic access. The robotic system enabled precise drainage in a patient otherwise unsuitable for less invasive alternatives [113].

Moreover, robotic-assisted techniques described for walled-off pancreatic necrosis (WOPN) may also be applicable to the management of PPCs. Hogen et al. reported a robotic approach combining transgastric cystogastrostomy with ultrasound-guided pancreatic debridement in a patient with WOPN, using a stapled anastomosis to shorten operative time and improve procedural efficiency. While the case involved necrotic collections, the authors emphasized that a similar robotic technique can be employed for PPCs, particularly when endoscopic drainage is not feasible or when a more controlled, surgically guided approach is desired [114].

Despite its advantages, robotic surgery for PPCs is still limited by several factors. These include longer operative times, higher costs, and the necessity for specialized surgical expertise [116].

Taken together, while robotic surgery is not yet standard practice in PPC management, growing clinical experience suggests that it can offer significant benefits in selected patients, particularly those with complex anatomy, large or infected PPCs, or those requiring adjunctive procedures. As technology and surgical expertise continue to evolve, the role of robotic techniques in pancreatic surgery is expected to expand.

### 7.2. Hybrid Method

In 2020, Dana and Adrian Bartos presented a hybrid surgical technique which consisted of endoscopy drainage assisted by single transgastric trocar, laparoscopic pseudocystogastrostomy. The new method was applied to three patients with 13–18 cm PPCs with repeated episodes of acute pancreatitis. The procedures were performed using intraoperative ultrasound. No complications were observed, and the method was described as facile and superior to only endoscopic drainage due to the wider connection between the stomach and the PPC, which reduces the risk of recurrence and the possibility of exploring the cavity and removing necrotic tissue [119].

## 8. What Is the Best Treatment for Pancreatic Pseudocysts?

Historically, in 1995, Catalano proposed an algorithm for the treatment of PPCs, which depended on the communication of the PPCs with the PD. When PPCs communicated with the PD and there was presence of strictures, dilatation and duct drainage with a stent was performed. However, without the presence of strictures, transpapillary cyst drainage with a ‘’J’’-shaped guidewire or pigtail was performed. On the other side of this algorithm, in the case of PPCs that do not communicate with the PD, endoscopic drainage was performed depending on the gastric or duodenal bulge, and in the absence of a bulge, a non-surgical option (radiological drainage) or a surgical option was left [78].

In 2015, a new system was developed to classify PPCs into different therapeutic solutions, which was based on PPC size, location, connection with the PD, and symptoms. Endoscopic drainage was then considered the optimal strategy [88].

Over time, due to its more pronounced advantages, endoscopic drainage has become the first-line method for treating PPCs until today compared with surgical and percutaneous drainage [1,120,121]. Advantages include the minimally invasive nature of the procedure, the possibility of placing more than one drain, irrigation of the cyst, and direct necrosectomy. Thanks to this method, the number of complications and the risk of recurrence have also decreased. It is also associated with shorter hospital stays and lower costs than surgical methods [1,121].

The newest guidelines from international expert panels—European, American, Korean, and Chinese—are summarized in Table 1. This table outlines key recommendations from leading gastroenterological and multidisciplinary societies regarding the management of PPCs, with a focus on the indications and preferred use of endoscopic, percutaneous, and surgical drainage techniques. Moreover, the summary of the literature reports on the management of PPCs are presented in Table 2.

## 9. Conclusions

The management of PPCs has undergone significant evolution over the past centuries. Historically, open surgical methods such as marsupialization, external drainage, and internal drainage procedures like cystogastrostomy and cystojejunostomy were the standard treatments. While effective, these approaches were associated with considerable morbidity and extended recovery periods. The advent of laparoscopic surgery introduced a less invasive alternative, offering benefits such as reduced postoperative pain and shorter hospital stays. Endoscopic drainage has further revolutionized treatment, emerging as the preferred modality for many patients. This approach allows for internal drainage of the PPC into the gastrointestinal tract without external incisions, resulting in comparable success rates to surgical methods but with reduced adverse events and shorter hospital stays. Looking ahead, robotic-assisted surgery presents an exciting frontier in the management of PPCs.

## Figures and Tables

**Figure 1 jcm-14-06152-f001:**
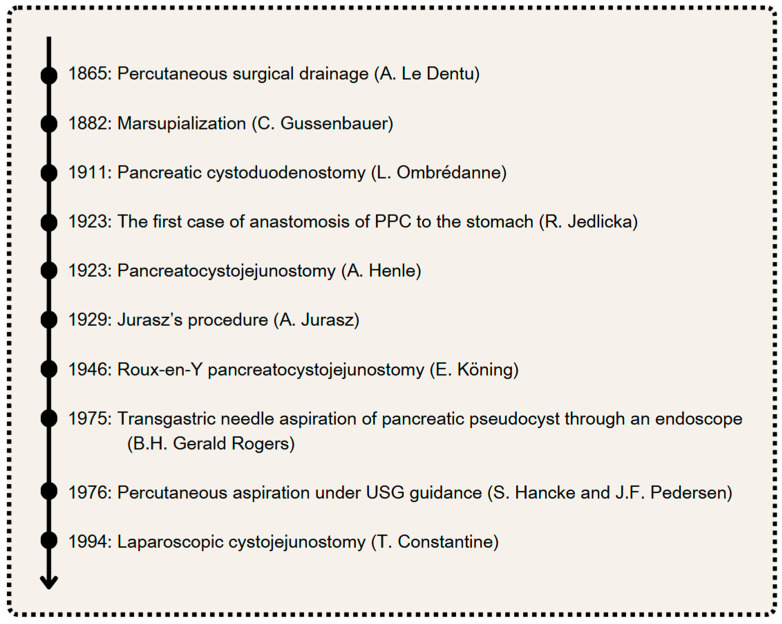
The most important dates in the history of PPC treatment. Author: Aleksandra Tobiasz.

**Figure 2 jcm-14-06152-f002:**
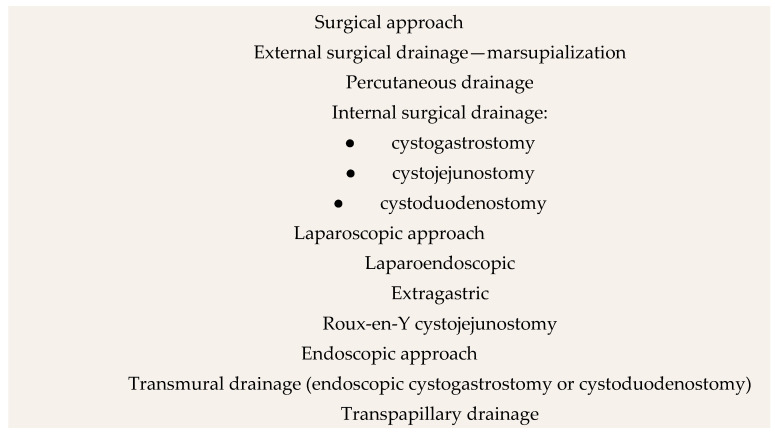
Interventional methods of treatment in PPCs. Author: Aleksandra Tobiasz.

**Figure 3 jcm-14-06152-f003:**
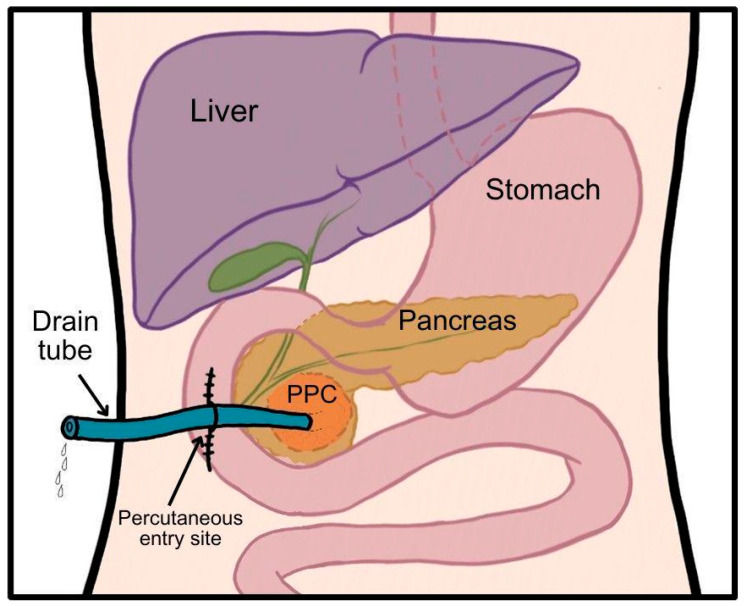
Schematic presentation of percutaneous drainage. Author: Aleksandra Tobiasz.

**Figure 4 jcm-14-06152-f004:**
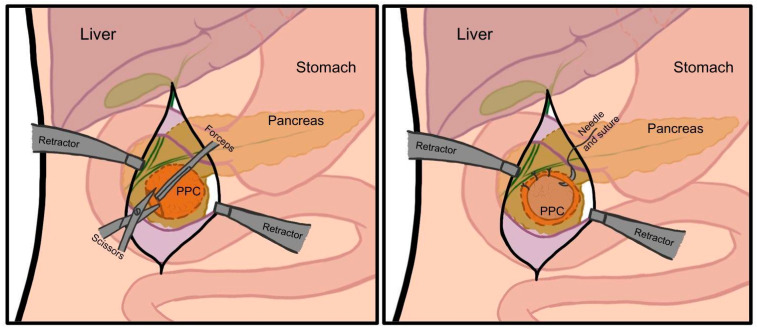
Schematic presentation of marsupialization. Author: Aleksandra Tobiasz.

**Figure 5 jcm-14-06152-f005:**
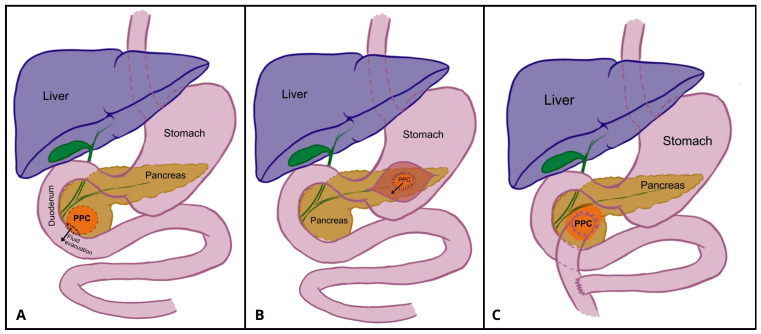
Schematic presentation of pancreatocystoduodenoostomy (**A**), pancreatocystgastrostomy (**B**), and Roux-en-Y pancreatocystojejunostomy (**C**). Author: Aleksandra Tobiasz.

**Figure 6 jcm-14-06152-f006:**
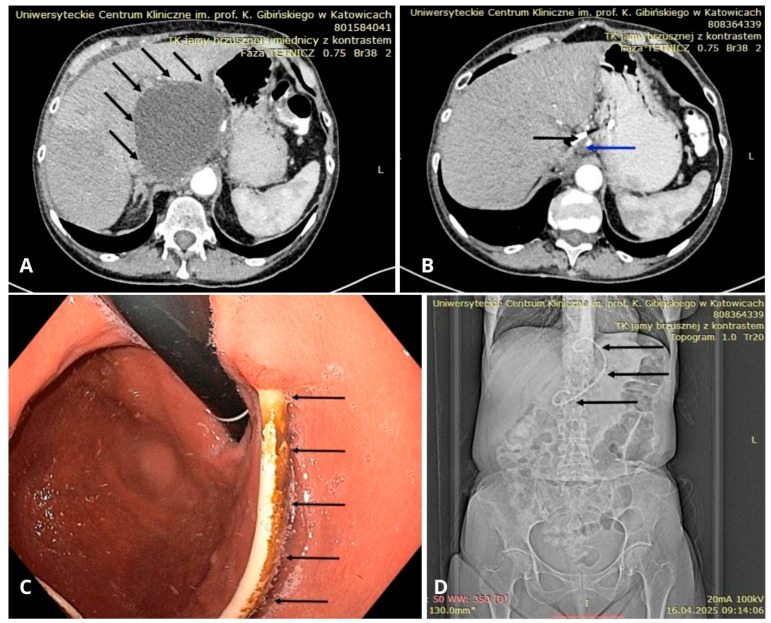
(**A**) CT scan showing a large peripancreatic pseudocyst (black arrows). (**B**) CT scan showing the residual pancreatic pseudocyst (blue arrow) from 3a after several weeks of drainage with a DPPS (black arrow). (**C**) Endoscopic view of the stomach showing a DPPS (black arrows) implanted through the gastric wall into the lumen of the pancreatic pseudocyst. (**D**) Follow-up abdominal CT scan after endoscopic drainage of the PPC—scout view showing a DPPS (black arrows). Source: Department of Gastroenterology and Hepatology and Endoscopy Department of the University Clinical Centre of the Prof. K. Gibliński in Katowice.

**Figure 7 jcm-14-06152-f007:**
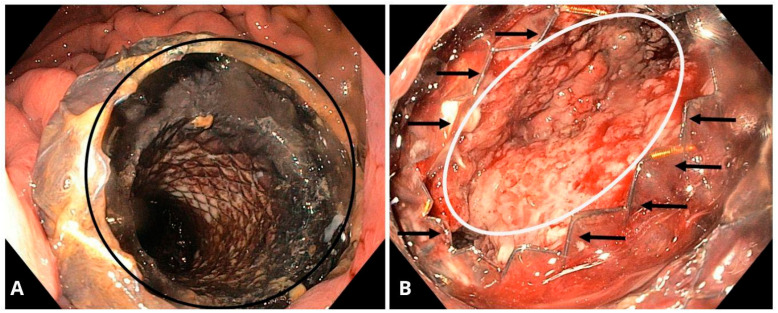
(**A**) Endoscopic view of the stomach showing a self-expanding, covered, removable metal stent (black circle) creating a connection between the gastric lumen and the pancreatic pseudocyst. (**B**) Endoscopic view of the cavity in the patient from 1a, showing the interior of the collection with advanced granulation tissue (white oval). The end of the metal stent is indicated by black arrows. Source: Department of Gastroenterology and Hepatology and Endoscopy Department of the University Clinical Centre of the Prof. K. Gibliński in Katowice.

**Figure 8 jcm-14-06152-f008:**
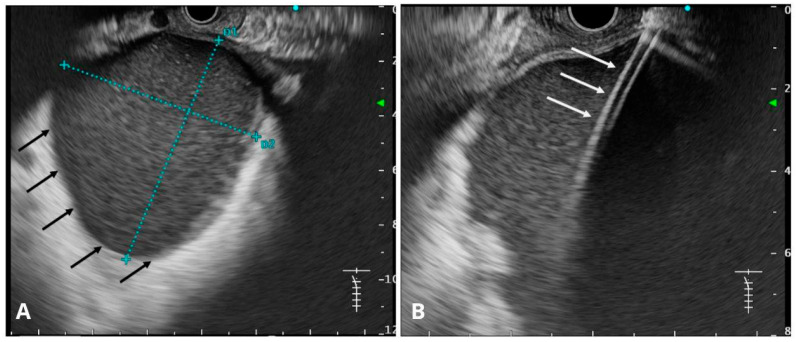
(**A**) Endosonographic image of the pancreatic pseudocyst (black arrows). The dimensions of the collection are marked with a blue dashed line. (**B**) Endosonographic image of a cystotome inserted into the pancreatic pseudocyst (white arrows). Source: Department of Gastroenterology and Hepatology and Endoscopy Department of the University Clinical Centre of the Prof. K. Gibliński in Katowice.

**Figure 9 jcm-14-06152-f009:**
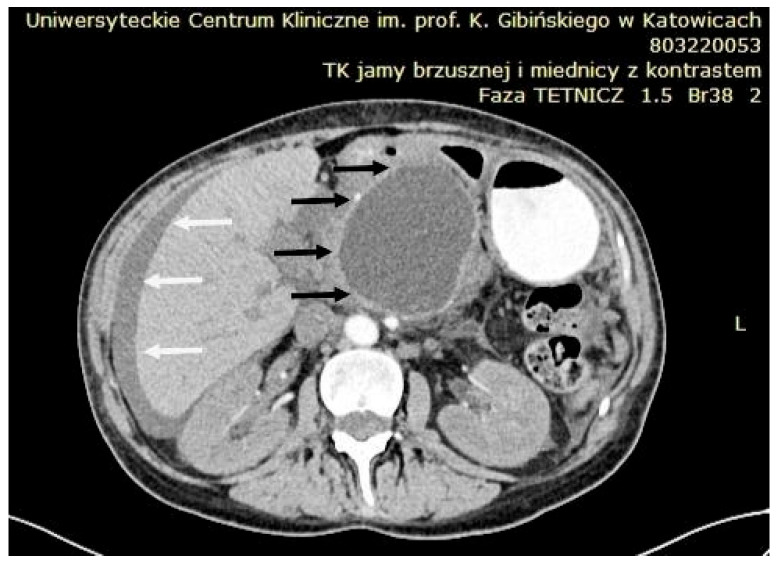
CT scan showing a large peripancreatic pseudocyst (black arrows) compressing the gastric wall. Associated ascites is indicated by white arrows. Source: Department of Gastroenterology and Hepatology and Endoscopy Department of the University Clinical Centre of the Prof. K. Gibliński in Katowice.

**Table 1 jcm-14-06152-t001:** Summary of major guideline recommendations for the management of pancreatic pseudocysts.

Guidelines	Conclusions
ESGE 2018 [122]	Endoscopic drainage is recommended over percutaneous or surgical approaches for uncomplicated chronic pancreatitis-related PPCs within endoscopic reach.
KSGE 2021 [123]	Endoscopic drainage is preferred for pancreatic fluid collection adjacent to the stomach or duodenum. If endoscopic access is not feasible, percutaneous drainage may be considered. Surgical intervention is reserved for cases unresponsive to endoscopic or percutaneous treatment or complicated by issues such as bleeding.
ASGE 2024 [121]	Endoscopic drainage is suggested over surgical drainage for symptomatic pseudocysts in patients with chronic pancreatitis.
Chinese expert panel 2024 [124]	US-guided puncture and drainage is the first-line treatment for PPCs. Roux-en-Y cystojejunostomy is recommended when the PPC is distant from the stomach/duodenum and percutaneous drainage is not feasible. If endoscopic and conservative treatments fail, laparoscopic internal drainage is preferred. Percutaneous catheter drainage is advised when the PPC is distant from the stomach wall but accessible percutaneously.

**Table 2 jcm-14-06152-t002:** Summary of the literature reports on the management of pancreatic pseudocysts.

Authors	Type of Study	Year	Patients No.	Surgery	Laparoscopy	Endoscopy	Percutaneous Drainage	Success Rate	Conclusion
R F Murphy et al. [26]	retrospective study	1960	35	X	-	-	X	-	Internal drainage is the most effective treatment; catheter drainage is preferable among external methods, with efficacy comparable to marsupialization.
E L Bradley [49]	retrospective study and review	1984	14	X	-	-	-	-	Transduodenal cystoduodenostomy shows comparable mortality to other internal drainage methods; laterolateral cystoduodenostomy is associated with high mortality and is not recommended.
V P O’Malley et al. [13]	retrospective study	1985	69	X	-	-	X	-	Surgery is effective but carries notable morbidity and recurrence, especially with infected PPCs.
J Sahel et al. [72]	retrospective study	1987	19	-	-	X	-	90%	Endoscopic cystic-digestive diversion offers a viable alternative to surgery in select cases with cysts compressing adjacent GI structures; careful patient selection and technique refinement are essential.
E van Sonnenberg et al. [22]	retrospective study	1989	101	-	-	-	X	90.1%	Percutaneous drainage is an effective front-line treatment for most PPCs.
M Cremer et al. [73]	retrospective study	1989	33	-	-	X	-	ECD vs. ECG (96% vs. 100%)	ECD is the preferred first-line treatment for paraduodenal cysts; ECG is effective for retrogastric pseudocysts, with PCD as adjunct in infection. Surgery is reserved for endoscopically inaccessible cases.
K A Newell et al. [43]	clinical Trial	1990	98	X	-	-	-	-	Cystgastrostomy and cystjejunostomy have similar outcomes; cystgastrostomy is preferred due to shorter operative time and reduced blood loss when anatomically feasible.
A D’Egidio et al. [27]	prospective study	1992	21	-	-	-	X	-	PCD is preferred for symptomatic, large, or expanding type I cysts and effective in type II cysts lacking ductal communication.
M F Catalano et al. [78]	retrospective study	1995	21	-	-	X	-	80.9%	Transpapillary pancreatic duct stenting is a safe and effective first-line treatment for symptomatic PPCs with ductal communication.
M E Smits et al. [80]	retrospective study	1995	37	-	-	X	-	-	Endoscopic drainage is feasible and safe and was a definitive treatment for two-thirds of the patients (65%); surgery is reserved for endoscopic failure.
M Barthet et al. [81]	clinical trial	1995	30	X	-	X	-	76%	ETCD is a safe and effective method for draining PPCs with ductal communication.
G C Vitale et al. [77]	retrospective study	1999	29	-	-	X	-	83%	Endoscopic drainage is a safe, effective, and often definitive treatment, and it should be considered before surgery in appropriately selected patients.
W Testi et al. [46]	retrospective study	2001	22	X	-	-	-	-	Roux-en-Y cystojejunostomy remains the first-line elective treatment for PPCs; alternative drainage methods (endoscopic internal drainage and surgical or percutaneous external drainage) should be reserved for complications or high surgical risk.
A E Park et al. [53]	retrospective study	2002	28	-	X	-	-	96%	Laparoscopic internal drainage is effective for large or endoscopy-inaccessible PPCs, offering fast recovery, low morbidity, and no mortality in selected patients.
M Cantasdemir et al. [5]	retrospective study	2003	30	-	-	-	X	96%	Percutaneous drainage is a safe and effective front-line treatment for patients with infected PPCs.
P Hauters et al. [66]	retrospective study	2004	17	-	X	-		94%	Laparoscopic treatment of PPCs is effective with low complication rates and avoids bleeding risks associated with endoscopic internal drainage.
D Cahen et al. [86]	retrospective study	2005	92	-	-	X	-	71%	Endoscopic drainage is effective in most cases; use of pigtail stents and proactive infection management may reduce complications.
L Weckman et al. [87]	retrospective study	2006	170	-	-	X	-	86.1%	Endoscopic treatment is safe and effective; surgery is reserved for inaccessible ducts, unfavorable PPC characteristics, or endoscopic complications.
C Palanivelu et al. [67]	retrospective study	2007	108	X	X	-		-	Laparoscopy is a safe, effective surgical option for pseudocysts, offering excellent long-term outcomes, short hospital stays, and early diet resumption.
M Arvanitakis et al. [90]	randomized controlled trial	2007	77	-	-	X		-	Stent retrieval after successful transmural drainage is linked to increased recurrence of pancreatic collections.
P J Talreja et al. [92]	prospective case series	2008	18	-	-	X	-	95%	CSEMS placement is a safe and effective option for PPC drainage; randomized trials are needed to compare with plastic stents.
S Varadarajulu et al. [109]	prospective randomized trial	2008	30	-	-	X		drainage by EUS vs. EGD (95.8% vs. 80%)	EUS-guided drainage should be first line for PPCs due to its high technical success rate.
L Melman et al. [65]	retrospective study	2009	83	X	X	X	-	surgery vs. laparoscopic vs. endoscopy (90.9% vs. 93.8% vs. 84.6%)	Laparoscopic and open cystogastrostomy have higher primary success than endoscopic drainage, though repeat endoscopy can achieve success in selected cases.
M D Johnson et al. [117]	retrospective study	2009	61	X	-	X	X	surgery vs. endoscopy (93.3% vs. 87.5%)	Surgical and endoscopic treatments are equally effective; endoscopic drainage is preferred as initial therapy, with percutaneous drainage playing a limited role.
N Hamza et al. [54]	retrospective study	2010	28	-	X	-	-	-	Laparoscopic drainage is highly effective with low morbidity, rapid recovery, and recurrence rates similar to open surgery; approach depends on PPC size and location.
R Z Sharaiha et al. [93]	retrospective cohort study	2015	230	-	-	X		DPPSs vs. FCSEMSs (89% vs. 98%)	EUS-guided drainage with FCSEMSs offers better clinical outcomes and fewer adverse events than DPPSs in PPC management.
R J Shah et al. [97]	clinical trial	2015	33	-	-	X	-	93%	LACSEMSs were successfully placed in 91% of pancreatic fluid collections cases, with 93% achieving resolution; advantages include single-step deployment, endoscopic debridement capability, and low migration rates.
G Pan el al. [88]	retrospective study	2015	893	X	-	X	X	surgery vs. endoscopy (93.3% vs. 88.9%)	Surgical and endoscopic treatments are effective and safe; endoscopic drainage is preferred first line in suitable patients due to its minimal invasiveness.
A A Redwan et al. [63]	multicenter retrospective study	2017	71	X	X	X	-	surgery vs. laparoscopy vs. endoscopy (100% vs. 100% vs. 82.9%)	The management of PPCs is evolving, with minimally invasive techniques emerging as effective alternatives to open surgery. Endoscopic methods show excellent outcomes, and laparoscopy is a promising but technically demanding option.
N Ge et al. [103]	retrospective study	2017	52	-	-	X	-	100%	Both plastic stents and LAMSs are effective for PPC drainage, but LAMSs offer advantages in reducing migration, cyst leakage, and need for reintervention.
Y Wang et al. [62]	retrospective study	2019	248	X	X	-	X	-	Laparoscopic drainage of PPCs is associated with fewer short-term complications and superior outcomes compared to percutaneous and open surgical approaches.
J Yang et al. [102]	multicenter, international retrospective study	2019	205	-	-	X		LAMSs vs. DPPSs (96.3% vs. 87.2%)	LAMSs are safe, effective, and superior to DPPSs for PPCs, offering higher clinical success, shorter procedures, fewer percutaneous interventions, and lower adverse event rates.
F Yetisir et al. [68]	retrospective study	2020	14	-	X	-	-	100%	Laparoscopic drainage remains the gold standard for suitable PPCs, offering highest success and lowest recurrence rates. Linear-stapled laparoscopic cystogastrostomy via anterior gastrostomy is effective and safe for retrogastric PPCs.
M Sarzamin et al. [48]	randomized clinical trial	2021	140	X	-	-	-	-	Cystogastrostomy is associated with higher postoperative recurrence rates compared to cystojejunostomy.
M V Marino et al. [114]	case-series retrospective study	2021	14	X	-	-	-	92.8%	Robotic drainage of symptomatic PPCs is safe, feasible, and a viable option in selected patients.
J Y Bang et al. [96]	retrospective, observational before–after study	2022	160	-	-	X		LAMSs with plastic stents vs. only plastic stents (95.6 vs. 89.4%)	A structured approach combining LAMSs with selective plastic stent use enhances treatment success for pancreatic fluid collections over plastic stents alone.
M E L D Santos et al. [89]	randomized clinical trial	2023	42	-	-	X		LAMS vs. SEMS (85.71% vs. 95.24%)	FCSEMS and LAMS show comparable efficacy and safety in EUS-guided drainage of EPCs; An FCSEMS offers shorter procedure time and fewer intra-procedure complications. Stent selection should consider availability, cost, and clinical expertise.
P Kluszczyk et al. [11]	retrospective study	2024	39	-	-	X		LAMSs vs. DPPSs (100% vs. 85.71%)	Both DPPSs and LAMSs demonstrate high therapeutic success and low complication rates in endoscopic drainage of PPCs.

Abbreviations for Table 2: X—present in the study, PCD—percutaneous drainage, ECG—endoscopic cystogastrostomy, ECD—endoscopic cystoduodenostomy, PPC—pancreatic pseudocyst, ETCD—endoscopic transpapillary cyst drainage, CSEMSs—covered, self-expanding metallic stents, DPPSs—double-pigtail plastic stents, FCSEMSs—fully covered, self-expanding metal stents, LAMS—lumen-apposing metal stent, LACSEMS—lumen-apposing, covered, self-expanding metal stent.

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
