# Peer review of "Pancreatic Pseudocysts: Evolution of Treatment Approaches"

_jcm, 2025, doi:10.3390/jcm14176152_

Round 1

Reviewer 1 Report

Comments and Suggestions for Authors

I appreciate the opportunity to review this interesting article. It is a review article that addresses the evolution of treatments for pancreatic pseudocysts. Overall, it is a detailed review with an extensive evaluation of the techniques used to treat patients with this condition. The scientific references are relevant and mention international treatment guidelines and recommendations. The images and tables included in the article are useful for providing a schematic overview. My recommendation is based on the article's length and relevance to clinical practice. While the information on historical development is relevant, I believe it could be better summarized as the article is currently very long and resembles a textbook chapter more than an article. To maintain the authors' focus on the evolution of treatment techniques, I suggest summarizing the general historical context and emphasizing the scientific and clinical basis of the practices recommended in the different international guidelines. Additionally, there should be a section that briefly addresses possible new therapies or opportunities for therapeutic development. This will make the information relevant to the format of a scientific publication, meaning the main focus should be on providing the most relevant information on the current therapy, using the historical link only as context for the current development of the techniques.

Author Response

Dear Reviewer,

Thank you for peer reviewing of our manuscript jcm-3784162, entitled "Pancreatic Pseudocysts: Evolution of Treatment Approaches”.

Thank you for your questions and comments. We have fully addressed all the comments and my responses appear below. Our revised work includes corrections according to reviewers’ comments in the text. The changes made according to reviewers’ comments are highlighted in yellow in the main text.

We take this opportunity to express my gratitude to the reviewers for their constructive and useful remarks. Their comments allowed us to identify areas in our manuscript that needed modification.

We also thank you for allowing me to resubmit a revised copy of the manuscript. We hope that the revised manuscript is now acceptable for publication in the Journal of Clinical Medicine.

Yours sincerely,

Paulina Kluszczyk, Aleksandra Tobiasz and Beata Jabłońska

Responses to Reviewer 1

Comment: My recommendation is based on the article's length and relevance to clinical practice. While the information on historical development is relevant, I believe it could be better summarized as the article is currently very long and resembles a textbook chapter more than an article. To maintain the authors' focus on the evolution of treatment techniques, I suggest summarizing the general historical context and emphasizing the scientific and clinical basis of the practices recommended in the different international guidelines.

Answer: Thank you for this comment. In accordance with your suggestion, we have summarized the manuscript, with particular attention to condensing the historical content in both the surgical and endoscopic sections (6.1; 6.2; 6.2.1; 6.3; 6.3.1). This allowed us to focus more on the clinical aspects and the scientific basis of current practices, while keeping the historical context concise.

Comment: Additionally, there should be a section that briefly addresses possible new therapies or opportunities for therapeutic development. This will make the information relevant to the format of a scientific publication, meaning the main focus should be on providing the most relevant information on the current therapy, using the historical link only as context for the current development of the techniques.

Answer:Thank you for the valuable comment. While the primary aim of the manuscript is to present a historical and practical overview of current treatments, we have expanded the section on robotic surgery to highlight it as a promising area for future therapeutic development.

Kind regards,

Paulina Kluszczyk, Aleksandra Tobiasz and Beata Jabłońska

Reviewer 2 Report

Comments and Suggestions for Authors

As we know, pancreatic pseudocysts (PPCs) are a fairly common complication of both acute and chronic pancreatitis. The paper mainly reviews current surgical treatments, especially open procedures, but doesn't really bring anything new in terms of breakthroughs—either in understanding the underlying mechanisms or identifying drug targets. It reads more like a summary of existing surgical practices than something with strong clinical relevance.

Also, the figures in the paper are quite basic and not very polished. It would be more effective to use composite figures rather than listing single images one after another.

Overall, the paper lacks originality, and the visuals don’t help much either. If the authors are considering a major revision, improving the quality and clarity of the figures might be a good place to start.

Comments on the Quality of English Language

Could be improved by experts in this area.

Author Response

Dear Reviewer,

Thank you for peer reviewing of our manuscript jcm-3784162, entitled "Pancreatic Pseudocysts: Evolution of Treatment Approaches”.

Thank you for your questions and comments. We have fully addressed all the comments and my responses appear below. Our revised work includes corrections according to reviewers’ comments in the text. The changes made according to reviewers’ comments are highlighted in yellow in the main text.

We take this opportunity to express my gratitude to the reviewers for their constructive and useful remarks. Their comments allowed us to identify areas in our manuscript that needed modification.

We also thank you for allowing me to resubmit a revised copy of the manuscript. We hope that the revised manuscript is now acceptable for publication in the Journal of Clinical Medicine.

Yours sincerely,

Paulina Kluszczyk, Aleksandra Tobiasz and Beata Jabłońska

Responses to Reviewer 2

Comment: As we know, pancreatic pseudocysts (PPCs) are a fairly common complication of both acute and chronic pancreatitis. The paper mainly reviews current surgical treatments, especially open procedures, but doesn't really bring anything new in terms of breakthroughs—either in understanding the underlying mechanisms or identifying drug targets. It reads more like a summary of existing surgical practices than something with strong clinical relevance.

Answer: Thank you for your insightful comment. We agree that the manuscript focuses primarily on established surgical approaches; however, our goal was to provide a comprehensive and historically grounded overview of treatment strategies for pancreatic pseudocysts. We believe this synthesis — linking classical techniques with current practices — offers practical value for clinicians, especially in settings where advanced interventions may not be readily available. We hope this contextual review will serve as a useful reference and complement ongoing research into novel therapies.

Comment: the figures in the paper are quite basic and not very polished. It would be more effective to use composite figures rather than listing single images one after another.

Answer: Thank you for the suggestion. Figures have been reorganized into composite images (for example Figures 5–7 are now 5A–C; Figures 8–10 are now 6A–D), which we hope improves clarity and presentation.

Kind regards,

Paulina Kluszczyk, Aleksandra Tobiasz and Beata Jabłońska

Reviewer 3 Report

Comments and Suggestions for Authors

This paper provides a comprehensive overview of the historical progression and therapeutic advancements in the treatment of pancreatic pseudocysts. However, I would like to point out that there are several errors in the paper, as well as some areas that need to be clarified.

  1. Figure 1 summarizes the evolution of treatment methods for pancreatic cysts in chronological order. However, the treatments from 1929 and 1946 are listed in reverse order and should be corrected.
  2. Figures 6 and 7 are not cited or discussed in the main text.
  3. Although the manuscript provides an extensive description of laparoscopic drainage for pancreatic pseudocysts, it would be beneficial to elaborate on the precise indications for this approach. Additionally, a detailed comparison with endoscopic drainage—highlighting the specific advantages of the laparoscopic method—would enhance the clarity and clinical relevance.
  4. In lines 393–399, the manuscript states that the endoscopic approach for pancreatic pseudocysts has shown promising results. However, it is unclear whether this refers to the EUS-guided approach or the blind (non-EUS-guided) technique. Clarification is needed to avoid ambiguity.
  5. Lines 451–452 suggest replacing the stent if a pancreatic pseudocyst persists beyond 4–6 weeks following endoscopic drainage. It would be valuable to discuss whether simple exchange is adequate or if upsizing the stent—for instance, to a lumen-apposing metal stent (LAMS)—should be considered to enhance drainage efficacy.
  6. In line 457, the authors state that lesions located within 1 cm of the gastrointestinal lumen are associated with a higher recurrence rate. Please provide specific evidence and references to support this statement."
  7. The manuscript lacks a description of therapeutic approaches for pancreatic pseudocysts complicated by disconnected duct syndrome. Given the clinical significance of this condition, a discussion of its management—such as long-term stenting, surgical options, or endoscopic interventions—should be included.
  8. In lines 577–582, the manuscript cites a 2007 study to report the low utilization rate of EUS prior to endoscopic drainage based on an international survey. However, this reference may not reflect current clinical practice. It is recommended that the authors incorporate more recent data to support this point.
  9. In line 621, the abbreviation 'PC' appears, but it is unclear what it stands for.
  10. The manuscript mentions that robotic surgery can be a useful option for the treatment of pancreatic pseudocysts when endoscopic or percutaneous approaches are not feasible. However, it would be beneficial to discuss the specific indications for robotic intervention in more detail.

Author Response

Dear Reviewer,

Thank you for peer reviewing of our manuscript jcm-3784162, entitled "Pancreatic Pseudocysts: Evolution of Treatment Approaches”.

Thank you for your questions and comments. We have fully addressed all the comments and my responses appear below. Our revised work includes corrections according to reviewers’ comments in the text. The changes made according to reviewers’ comments are highlighted in yellow in the main text.

We take this opportunity to express my gratitude to the reviewers for their constructive and useful remarks. Their comments allowed us to identify areas in our manuscript that needed modification.

We also thank you for allowing me to resubmit a revised copy of the manuscript. We hope that the revised manuscript is now acceptable for publication in the Journal of Clinical Medicine.

Yours sincerely,

Paulina Kluszczyk, Aleksandra Tobiasz and Beata Jabłońska

Responses to Reviewer 3

Comment 1: Figure 1 summarizes the evolution of treatment methods for pancreatic cysts in chronological order. However, the treatments from 1929 and 1946 are listed in reverse order and should be corrected.

Answer: Thank you for the observation. The order has been corrected accordingly.

Comment 2: Figures 6 and 7 are not cited or discussed in the main text.

Answer: Thank you for the comment. Figures 6 and 7 have been merged into a single image (now Figures 5A and 5B) and are now cited and discussed in the main text.

Comment 3: Although the manuscript provides an extensive description of laparoscopic drainage for pancreatic pseudocysts, it would be beneficial to elaborate on the precise indications for this approach. Additionally, a detailed comparison with endoscopic drainage—highlighting the specific advantages of the laparoscopic method—would enhance the clarity and clinical relevance.

Answer: Thank you for this suggestion. We have summarised the general indications for laparoscopic surgery in this section:

Current evidence suggests that endoscopic approaches may be more suitable for managing chronic PPCs within the head and body of the gland, while laparoscopic interventions are better suited for acute PPCs, particularly those associated with necrotizing pancreatitis due to remove more necrotic tissue from the cyst while exploring the cyst wall structure than endoscopic drainage.

The use of laparoscopy is limited by incomplete anastomosis, gastric perforation or abdominal contamination. Nevertheless, laparoscopic surgery of the pancreas is becoming an increasingly popular method of treating mature PPC, as it is minimally invasive and provides effective drainage. [55,64,65,68].

Specific indications for each type of laparoscopy are listed in the paragraph corresponding to the method.

We also compared both methods – endoscopic and laparoscopic – as suggested. However, the reports do not provide information on the significant advantage of the laparoscopic method over the endoscopic method. We decided to add this fragment to the manuscript:

However, recent comparative studies have questioned whether laparoscopic drainage offers a distinct advantage over endoscopic or open surgical methods in terms of long-term success, morbidity, and mortality. Some reports have suggested minimal differences between these modalities, though minor benefits of endoscopic techniques—such as shorter operative time, reduced blood loss, and briefer hospitalization—have been noted [63,64,65].

A first meta-analysis (from 2021) compared a group of patients treated with endoscopic and laparoscopic methods and presented the following results: treatment success rate (50-100% vs. 78.9-100%) and overall success rate (89.2% vs. 88.8%), showing that there were no differences in treatment rates between the two groups. In addition, the total rate of adverse events was compared, which was 11.35% in the endoscopic group and 14.66% in the laparoscopic group. The rates ranged from 0-21.4% and 8.3-26.3%, respectively. No significant differences were observed here either.

However, significant differences were found in the duration of surgery and hospitalisation, showing that they were significantly shorter and with less blood loss in the endoscopic group than in the laparoscopic group. Additional analyses revealed that there was no difference in recurrence rates between the two groups [64].

Comment 4: In lines 393–399, the manuscript states that the endoscopic approach for pancreatic pseudocysts has shown promising results. However, it is unclear whether this refers to the EUS-guided approach or the blind (non-EUS-guided) technique. Clarification is needed to avoid ambiguity.

Answer: Thank you for this observation. According to this comment we decided to add information about fluoroscopic control (non-EUS-guided).

Comment 5: Lines 451–452 [469] suggest replacing the stent if a pancreatic pseudocyst persists beyond 4–6 weeks following endoscopic drainage. It would be valuable to discuss whether simple exchange is adequate or if upsizing the stent—for instance, to a lumen-apposing metal stent (LAMS)—should be considered to enhance drainage efficacy.

Answer: Thank you for this comment. We decided to add additional information to our manuscript, which we have compiled below:

However, subsequent multivariate analyses identified that implantation of multiple endoprostheses (instead of single stent placement) and drainage duration longer than 6 weeks predict a more favourable outcome of endoscopic drainage. The placement of more than one stent provides a wider drainage opening and a lower probability of stent occlusion, which appears to result in better drainage. [86].

More information about this topic is provided in paragraph 6.6. Timing of Stent Removal.

Comment 6: In line 457, the authors state that lesions located within 1 cm of the gastrointestinal lumen are associated with a higher recurrence rate. Please provide specific evidence and references to support this statement."

Answer: Thank you for this observation. We made changes to this paragraph, because the recurrence rate was not directly related to these factors. It was related to technical success. We supported all information with appropriate bibliography. We decided to add this information to our manuscript, which we have compiled below:

Additionally, technical success rates have been associated with:

  • PPCs located in the head or body of the pancreas (compared to the tail)
  • Lesions <1 cm from the gastrointestinal lumen
  • Pseudocysts related to chronic or necrotizing pancreatitis rather than acute presentations [60].

The recurrence rate for transmural drainage is estimated at around 5% [82], while for both methods combined (transmural and transpapillary drainage) it is around 8-9%[88].

Comment 7: The manuscript lacks a description of therapeutic approaches for pancreatic pseudocysts complicated by disconnected duct syndrome. Given the clinical significance of this condition, a discussion of its management—such as long-term stenting, surgical options, or endoscopic interventions—should be included.

Answer: Thank you for the comment. We acknowledge the clinical importance of pancreatic pseudocysts complicated by disconnected duct syndrome; however, this specific scenario falls outside the scope of our review, which focuses on general treatment approaches rather than complex or rare complications.

Comment 8: In lines 577–582 [598], the manuscript cites a 2007 study to report the low utilization rate of EUS prior to endoscopic drainage based on an international survey. However, this reference may not reflect current clinical practice. It is recommended that the authors incorporate more recent data to support this point.

Answer:Thank you for the suggestion. We decided to add additional this information to our manuscript, which we have compiled below:

Over the past decade, EUS-guided endoscopic transmural drainage has become the first-line treatment approach for managing persistent and symptomatic PPCs, including both sterile or infected variants. When compared with percutaneous or surgical drainage modalities, EUS-guided demonstrates comparable technical success, while offering a significantly lower incidence of procedural morbidity, reduced need for repeat interventions, and a shorter duration of inpatient care [101].

Comment 9: In line 621 [643], the abbreviation 'PC' appears, but it is unclear what it stands for.

Answer: Thank you for pointing this out. The abbreviation has been corrected to "PPC" as intended.

Comment 10: The manuscript mentions that robotic surgery can be a useful option for the treatment of pancreatic pseudocysts when endoscopic or percutaneous approaches are not feasible. However, it would be beneficial to discuss the specific indications for robotic intervention in more detail.

Answer: Thank you for the suggestion. We have expanded the relevant section to provide more detail on the specific indications for robotic intervention.

Kind regards,

Paulina Kluszczyk, Aleksandra Tobiasz and Beata Jabłońska
